# Consequences of Structural Urbanism: Urban–Rural Differences in Cancer Patients’ Use and Perceived Importance of Supportive Care Services from a 2017–2018 Midwestern Survey

**DOI:** 10.3390/ijerph19063405

**Published:** 2022-03-14

**Authors:** Marquita W. Lewis-Thames, Patricia Fank, Michelle Gates, Kathy Robinson, Kristin Delfino, Zachary Paquin, Aaron T. Seaman, Yamilé Molina

**Affiliations:** 1Department of Medical Social Science, Center for Community Health, Northwestern University Feinberg School of Medicine, Chicago, IL 60611, USA; 2Department of Psychiatry, and Behavioral Sciences, Rush University Medical Center, Chicago, IL 60612, USA; patricia_fank@rush.edu; 3Truman Medical Centers, University Health, Kansas City, MO 64108, USA; michelle.gates@uhkc.org; 4Simmons Cancer Institute, Southern Illinois University School of Medicine, Springfield, IL 62702, USA; krobinson@siumed.edu; 5Department of Surgery, Southern Illinois University School of Medicine, Springfield, IL 62702, USA; kdelfino84@siumed.edu; 6Spectrum Health, 100 Michigan St. NE, Grand Rapids, MI 49503, USA; paquinzw@gmail.com; 7Department of Internal Medicine, University of Iowa, Iowa City, IA 52242, USA; aaron-seaman@uiowa.edu; 8Holden Comprehensive Cancer Center, University of Iowa, Iowa City, IA 52242, USA; 9Division of Community Health Sciences, University of Illinois at Chicago, Chicago, IL 60612, USA; ymolin2@uic.edu; 10University of Illinois Cancer Center, Chicago, IL 60612, USA

**Keywords:** cancer survivors, rural health, health services underuse, healthcare disparities, healthcare utilization

## Abstract

Background: Structural inequities, in part, undergird urban–rural differences in cancer care. The current study aims to understand the potential consequences of structural inequities on rural and urban cancer patients’ access to and perceived importance of supportive cancer care resources. Methods: We used data collected from November 2017 to May 2018 from a larger cross-sectional needs assessment about patients’ support needs, use of services, and perceptions at a Midwestern United States cancer center. Oncology patients received a study packet during their outpatient clinic visit, and interested patients consented and completed the questionnaires. Results: Among the sample of 326 patients, 27% of the sample was rural. In adjusted logistic regression models, rural patients were less likely to report using any secondary support services (15% vs. 27%; OR = 0.43, 95%CI [0.22, 0.85], *p* = 0.02) and less likely than urban counterparts to perceive secondary support services as very important (51% vs. 64%; OR = 0.57, 95%CI [0.33, 0.94], *p* = 0.03). Conclusion: Structural inequities likely have implications on the reduced access to and importance of supportive care services observed for rural cancer patients. To eliminate persistent urban–rural disparities in cancer care, rural residents must have programs and policies that address cancer care and structural inequities.

## 1. Introduction

Urban–rural cancer disparities are partly rooted in structural inequities that affect access to and utilization of quality cancer care resources. One example, Probst, Eberth and Crouch [1], describe Structural Urbanism as the practices and policies in the public health and healthcare systems that reinforce maintaining and rebuilding infrastructure to respond to a critical mass of payers. A consequence of Structural Urbanism is biased funding towards heavily populated healthcare centers and inequitable access to public health and healthcare resources in less populated rural areas (e.g., health professional shortage areas, medically underserved areas). Financial and political practices that support Structural Urbanism undergirds the closure of over 160 rural hospitals for over 15 years; thus, reducing the access that rural residents have to cancer prevention and control resources and contributing to widening urban–rural cancer disparities [2]. Reduced access to cancer care resources contributes to the burden of unmet cancer care needs by many rural cancer patients and their caregivers [3,4,5,6,7]. Structural inequities can also limit a patient’s cancer information gathering of important health information, consequently limiting the perceived importance and understanding of their cancer care needs.

### 1.1. Supportive Care for Cancer Patients

Comprehensive cancer care increasingly includes accessing services and resources that address a patient’s complex physical, psychological, spiritual, social, and informational support needs in addition to diagnostic and primary treatment [8,9,10,11]. Accordingly, national organizations such as the American Society of Clinical Oncology (ASCO) and the National Comprehensive Cancer Network (NCCN) have emphasized the importance of supportive care needs resources and established guidelines to assess and address these needs as part of a comprehensive care plan [12,13,14,15]. Hui and Bruera [11] provided three classifications to guide the conceptualization of supportive care. Primary supportive care is provided by primary cancer care teams (e.g., oncologists, nurses); involves active management of treatment-related adverse effects; and is often integrated into patients’ active treatment plans/visits with their primary care team. Secondary supportive care involves providers who specialize in one or more supportive care domains (e.g., psychiatrists, social workers, physical therapists). Access to secondary supportive care may require referrals and can be delivered in outpatient or inpatient settings. Tertiary supportive care involves specialists who regularly provide complex supportive care (e.g., palliative medicine) to referred patients in tertiary settings.

### 1.2. Urban–Rural Disparities in Accessing Supportive Cancer Care Services

Due to Structural Urbanism, rural cancer patients experience reduced access to resources (e.g., financial counselors, mental health professionals) that could resolve unmet supportive care needs (e.g., financial hardship, anxiety related to cancer) than their urban counterparts [16,17,18,19,20,21,22,23,24,25,26,27]. To access these services, rural residents often have to travel to distal urban centers, potentially bearing additional financial, social, and physical burdens to access these services [5,28]. This is unfortunate, as secondary supportive care services are available before, during, and after active treatment to minimize long-term unmet supportive care needs. Limitations to accessing supportive care need services lend to rural cancer disparities and unmet care needs that can persist as rural patients receive their care throughout their entire cancer journey.

Data quantifying urban–rural disparities in support service use, especially in terms of secondary services, remain limited, while extant research on supportive care use largely describes primarily urban samples (e.g., ref. [29]). Urban–rural disparities in secondary support services use may further be particularly striking, given secondary services are not well-integrated into patients’ visits; therefore, potentially unknown by cancer patients in less-resourced areas. Specifically, secondary support services are not routinely offered by the primary cancer care team. Structural Urbanism’s impacts on the access of services may thus translate to unequal knowledge about service benefits—leading to additional psychosocial and behavioral barriers to service utilization. Rural patients may underutilize secondary support services to a greater extent than urban cancer patients, given they are often in medically underserved and health professional shortage areas wherein evidence-based information is not as available. Advances in telemedicine and other promising solutions have the potential to reduce geographic and economic barriers to care [22]. These promising solutions may have limited utility in eliminating urban–rural disparities in preventable/manageable support needs, as they do not address the psychosocial consequences of Structural Urbanism (i.e., perceived importance of services). Urban–rural differences in the use of these services are crucial to investigate, given their role in widening disparities in rural–urban cancer outcomes and relatedness to persistent structural inequities. However, little research has quantified urban–rural differences in the perceived importance of supportive care.

### 1.3. Current Study

In summary, structural inequities potentially worsen cancer outcomes because they reduce access to cancer care resources, and in turn, limit the available cancer education resources that navigate patients to resources that provide quality cancer care. To understand the implications of structural inequities on cancer outcomes, it is important to examine the differences in cancer care resource use and its importance in urban and rural communities. To address this need, a Midwestern sample of urban and rural cancer patients seeking care at the same cancer center was studied. The hypotheses are provided below.

Relative to urban counterparts, fewer rural cancer patients will report secondary supportive services use.

Relative to urban counterparts, fewer rural cancer patients will perceive secondary supportive services to be important.

## 2. Methods and Materials

### 2.1. Design

The current study draws from a larger cross-sectional needs assessment about patients’ support needs, use of services, and perceptions at an academic-affiliated cancer center in the Midwestern United States. The Cancer Center’s catchment serves a large 82 county, of which 25 are urban counties, and the closest academic affiliated cancer center 85 miles away. All data were collected between November 2017 and May 2018. All procedures and materials were approved by the university’s institutional review board.

### 2.2. Procedures

Study investigators first identified patients from daily electronic clinic lists of patients who had scheduled outpatient appointments with a medical or surgical oncologist. Study packets were distributed after patients checked in for their scheduled appointments. The study packet included paper copies of consent forms and questionnaires. Interested patients provided written consent and completed questionnaires while awaiting their appointments. Only data from participants who signed the consent form were included in the final datasets. Incentives were not provided for study participation.

### 2.3. Inclusion Criteria and Study Sample

The larger needs assessment focused on the healthcare needs and the preferences and experiences of all patients seen by hematology and oncology providers, including those with and without confirmed cancer diagnoses. Inclusion criteria for the current study were: (1) being at least 18 years of age, (2) speaking and reading English, (3) a documented cancer diagnosis, and (4) available residential addresses. Based on these criteria, 161 participants who did not have a confirmed cancer diagnosis and 4 participants who were missing residential addresses were excluded.

### 2.4. Exclusion Criteria

Prisoners were not eligible for participation per institutional review board regulations.

### 2.5. Measures

***Rurality (Primary Predictor).*** We used Rural–Urban Continuum Codes (RUCC) affiliated with patients’ addresses in the medical records [30]. RUCC codes distinguish urban and rural counties in terms of population size, degree of urbanization, and adjacency to urban areas. For the current study, in line with standard procedures and given our frequency distribution (Table 1), we classified counties with RUCC codes of 4–9 as rural, which includes counties with populations ranging from less than 2500 to 20,000 or more and counties that may or may not be adjacent to urban areas.

***Secondary Support Services Use (Outcome).*** Patients were asked to report their use of support services offered at the specific cancer center (No, Yes). For the current study, 7 services (see Appendix A) were identified as helpful across different cancer diagnoses and types of treatment were the primary focus. The services we use are helpful in the context of this pilot study—however, we do not consider the breadth of these services extensive and representative of all potential supportive services. For primary inferential analyses, a summary dichotomous variable was created to indicate whether patients reported using any support services (None, Any).

***Perceived Importance of Secondary Support Services (Outcome).*** Patients rated the perceived importance of 14 services (see Appendix A) on a 4-point Likert-type scale (0 = Not important at all, 1 = A little important, 2 = Quite important, 3 = Very important). An aim of the larger study was to assess patients’ perspectives on these services and inform future planning for additional services. However, these services were not currently offered to patients. Based on frequency distributions, variables were dichotomized (Not important at all/A little Important = Not Important; Quite important/Very important = Very Important), as shown in Table 2. For primary inferential analyses, a summary dichotomous variable was created to indicate whether patients reported any of the potential support services to be quite important or very important (None, Any).

***Demographic and Cancer-related Factors (Covariates).*** The following information from patients’ medical records was abstracted: age (years); sex (male, female); race/ethnicity (non-Hispanic White, Other); marital status (married, not married); type of primary insurance (none, public, private); type of cancer diagnosis; treatment status (active treatment (i.e., currently undergoing treatment), not active); and treatment type they had received before study participation (surgery, radiation, chemotherapy; single or multiple treatments). Based on non-normal distributions, age was reclassified into a 4-category ordinal variable (18–56, 57–65, 66–72, 73+ years old). Regarding cancer diagnosis, based on preliminary frequency distributions (Table 1), we classified cancer diagnoses as reproductive (breast and genital cancers) or non-reproductive.

***Severe Support Needs (Covariate).*** The Symptoms and Concerns Checklist, a 29-item self-report measure of symptoms and concerns, which has been validated for use with cancer patients, was administered [31,32]. Respondents rated their degree of difficulty with different support needs on a 4-point Likert-type scale (0 = Not at all, 1 = A little, 2 = Quite a bit, 3 = Very much). Consistent with this instrument’s scoring protocols, to identify severe support needs, the 29 items were dichotomized (Not at all/A little = Not severe, Quite a bit/Very much = Severe) and summed (range: 0–29).

### 2.6. Analytic Plan

All analyses were conducted in SPSS 25 (IBM SPSS Statistics for Windows, Armonk, NY, USA). With regard to missing data, we used whole case analysis, wherein we excluded 4 patients who were missing residential addresses to determine their urban/rural status. First, descriptive statistics were assessed, and unadjusted bivariate analyses were conducted to assess crude urban–rural differences, including regression (age, # of severe support needs) and chi-square tests (sex, race, marital status, insurance status, cancer diagnosis, treatment status, treatment type, and multiple treatments specific severe support needs, any support service use, specific services use, importance of any specific services; importance of specific services; Table 1 and Appendix A). Second, analyses to examine associations between covariates and outcome variables were also conducted (Table 2). Third, adjusted logistic regression models were conducted to examine urban–rural differences in secondary support services use and perceived importance of secondary support services. We compared model fit across models that adjusted for different domains of covariates (age; demographic factors; socioeconomic factors; cancer-related factors; final model including factors with significant associations with outcome variables; Table 3). In models with demographic factors, variables included were: age (continuous), sex (male, female), race (non-Hispanic White, Other), marital status (married, not married), and insurance (private insurance, other). In models with cancer-related factors, the following variables were included: type of cancer diagnosis (reproductive, not reproductive), severe support needs (continuous), surgery (no, yes), radiation (no, yes), chemotherapy (no, yes), multiple treatments (no, yes), and active treatment status (no, yes). Due to sample size, models including all covariates were unable to be conducted. Given this, demographic and cancer-related factors that had shown significant associations with predictors and outcomes in terms of crude models (Table 1 and Table 2) were included in ‘final’ models. Likelihood ratios to compare model fit between age-adjusted models and models adjusting for multiple covariates were reported.

## 3. Results

In total 1214, patients received a survey (See Appendix A). Table 1 depicts urban–rural differences in demographic and cancer-related factors within the sample (*n* = 326). Among our urban sample, 83% lived in metro counties with <250,000 residents. Among our rural sample, 60% lived in counties with 2500–19,999 residents that were adjacent to a metropolitan area. Approximately 27% of the sample was rural, 22% was 73+ years old, 56% of the sample were women, 82% were non-Hispanic White, 59% were married, and 36% had private insurance. With regard to cancer-related factors, approximately 21% of the sample were diagnosed with a genital cancer, 42% were undergoing active treatment at the point of survey completion, 68% underwent surgery, and 63% had undergone multiple types of cancer treatment. Approximately 29% of the sample reported 5+ severe support needs. The most common unmet support needs concerned work, weakness, and intimacy. There were no significant urban–rural differences in demographic factors, cancer-related factors, and the number of severe support needs in our sample.

Table 2 describes the relationship between sample characteristics with support service use and the perceived importance of support services. About 24% of the sample had used at least one support service. Approximately 61% of the sample perceived at least one of the support services to be very important. Patients who used services were more likely to be younger (18–56: 24%, 57–65: 38%, 66–72: 16%, 73+: 15%), be in active treatment (33% vs. 17%), have undergone chemotherapy (29% vs. 19%), and have multiple treatments (30% vs. 13%). Patients that considered supportive services important were more likely to be younger (18–56: 72%, 57–65: 70%, 66–72: 48%, 73: 49%), have private insurance (71% vs. 55%), be in chemotherapy (66% vs. 55%), and have more support needs (5+ needs: 75%, 2–4 needs: 65%, 0–1 needs: 49%). Appendix A offers descriptive differences regarding urban–rural differences in use and the perceived importance of specific support services.

Table 3 depicts urban–rural associations of demographic, cancer-related, and severe support needs with patients’ use of support services and perceived importance of services. Relative to their counterparts, patients with cancer-related conditions were more likely to report using supportive services and a perceived importance for using supportive services. Urban–rural differences remained significant across all adjusted models for support service use (ORs = 0.42–0.48, ps = 0.01–0.03). Similarly, urban–rural differences in perceived importance of support services were largely significant across models (ORs = 0.52–0.60, ps = 0.02–0.04), except when including demographic covariates (OR = 0.63, 95%CI = 0.38, 1.05, *p* = 0.08). Adjusted models that incorporated cancer-related variables and ‘final’ models that incorporated demographic and cancer-related variables appeared to exhibit better fit than models which adjusted only for age (LRs = 16.44–34.88, ps = 0.01–<0.001).

## 4. Discussion

This study provides several important contributions to the emerging body of literature investigating the effects of structural inequities on urban–rural cancer care disparities [3,4,17,22,33]. First, relative to urban patients, rural patients were less likely to perceive the importance of secondary support services after adjusting for age, demographic characteristics, and cancer-related factors. This is a crucial finding, as it clarifies that patients’ understanding and perceived importance of services, not just their access to them, may partially explain urban–rural differences in secondary support service uptake. Patients’ perceived importance, however, must be contextualized in the landscape of Structural Urbanism—thus, patients may not perceive the benefits of these services due to a systemic lack of resources and access to evidence-based information. Second, urban–rural disparities in the use of secondary support services were quantified: Rural patients reported using any secondary support service less often than urban patients. This finding highlights the impact of known geographical and structural rural–urban disparities in accessing supportive care services even for patients that attend the same urban hospital.

A smaller proportion of rural patients perceived supportive care to be more important relative to urban patients. Multiple, potential concurrent, explanations may underlie these differences in perceived importance. First, some research has suggested that rural patients may be less likely to use cancer treatment options that are more time-intensive (e.g., multiple visits within shorter periods of time) [34,35,36,37]. Given the more limited time they may have with rural patients, providers may be less likely to discuss and coordinate ‘ancillary’ services focused on prevention and early detection of manageable support needs. Second, geographic and economic barriers may not only affect overall access to treatment but likely influence rural patients’ decision-making processes for care. Cancer patients with more intensive treatment regimens may require greater travel and more associated expenses (e.g., temporary housing near facilities, gas costs), which is particularly burdensome for rural residents with transportation and financial barrier. Considering these barriers, rural cancer patients may prioritize treatment options that are less time-intensive (e.g., extensive surgery) while compromising their recommended care. In line with this scenario, fewer rural patients may perceive secondary support services to be as important as overcoming barriers associated with the travel costs and healthcare fees that would be associated with their use. Notably, rural areas are rapidly aging, but rural residents are less likely to have ADA-friendly homes and paratransit services [38,39]. This can make multiple treatment visits or multiple visits to access supportive care services challenging and potentially dangerous. Finally, rural communities’ values and norms regarding autonomy and self-reliance [20,25] may also affect patients’ perceptions about secondary support services, especially when needs have not emerged yet or are not severe. Most likely, these multi-faceted factors, often associated with the effect of structural inequities, intersect and additively contribute to rural patients’ lower perceived importance of secondary support services, which, in turn, may affect their use of such services.

Fewer rural patients in this study reported using any secondary support services relative to urban patients. This particular finding is a likely consequence of structural inequalities in urban–rural differences in access to supportive care services. Numerous studies support our finding that rural cancer patients are less likely to use supportive care services and have higher unmet supportive care needs, but these examinations are often outside of the United States, where free or universal health coverage relieves some of the barriers associated with structural inequities (e.g., [5,40,41]). Uniquely, our findings underscore the burdens felt by rural residents in a country with structural inequities and likely burdened by healthcare costs. Policies and financial reform that facilitate additional aid to rural hospitals, restructure “per-patient” types of funding, and increase broadband internet for rural residents to access telemedicine and specialty clinics are a few suggestions to combat structural inequities and eliminate urban–rural cancer care disparities. There is also a need for additional investigations that describe associations (e.g., mediating and moderating effects) between structural factors such as insurance coverage or broadband access on urban–rural differences of cancer care and related outcomes in the United States.

It is well documented that cancer prevention and risk education resources tailored for rural residents are limited [41,42,43,44]. Our findings also support that supportive care needs education and programming which addresses the structural inequalities affecting rural cancer patients and their families are needed [45]. In the absence of trusted and reliable cancer education resources, rural residents may use less reliable information sources including the internet, social media, tobacco companies, or ill-informed friends and family [28,46]. In an investigation testing the feasibility of a program that provides a survivorship care plan with enhanced patient education resources compared to a standard survivorship care plan, participants with the enhanced survivorship care plan reported improvements in multiple supportive care needs including emotional support, physical well-being, and nutrition [47]. Similarly, cancer education resources tailored for rural residents can improve patients’ understanding of comprehensive cancer care. It is important to note that while supplemental cancer education is a potential cost-efficient strategy, we are not suggesting it is a suitable alternative to mitigate shortages in rural practice-based care.

## 5. Limitations

This study had several limitations that should be kept in mind when interpreting findings. Primarily, our analysis was limited by a smaller representation of rural residents, racially and ethnically diverse patients, and patients with multiple cancer types. The smaller sample of rural residents posed challenges to analyzing urban–rural differences in supportive services use. While the sample population was diverse in some respects (gender, age, diagnosis), it was homogeneous in other respects (race/ethnicity). Also related to the small sample size of rural residents, we were unable to include all covariates in the models into inferential analyses. Future studies should oversample rural and racial/ethnic diverse residents to improve generalizability and enable a detailed analysis of urban–rural differences. In addition, an analysis exploring patterns of utilization and prioritization of support services to assess urban–rural disparities in site-specific supportive services is warranted. Second, this sample only included English-speaking patients recruited during a scheduled appointment at the urban-based cancer center and was willing to complete a survey. Relatedly, the cancer center provides some supportive care services including cancer-specific and caregiver support groups, physical activity classes, and a wig salon; however, all of the services are offered within the primary urban setting. Altogether, study findings are limited by a selection bias and may only be applicable for patients able to read English and receiving care from an urban-based cancer center that address the supportive care needs of urban and rural patients. Third, certain cancer-related factors, education, and income were not collected for the parent study and thus were unable to be included in our analyses. The cancer-related variables are limited and not intended to be a comprehensive list of variables. Future studies that build on our results should incorporate additional cancer-related factors (e.g., duration of disease, prognosis) to make conclusions about the supportive services used at varying points of cancer care, and income and education, given well-known urban–rural socioeconomic disparities [48]. Fourth, self-report data were used to gather information on the use of support services, although medical record data were used to gather other cancer-related factors. Consequently, this outcome may have been affected by recall bias, social desirability bias, and other biases. Fifth, our measurement of perceived importance was not based on a previously validated instrument, which is warranted in future studies. Finally, the current study operationalized rurality in terms of RUCC codes, which are based on county-level designations and are likely to be less precise (e.g., over- or under-bounding) than more sophisticated geospatial operationalizations [48]. Future research should confirm our findings with alternative, more comprehensive measurements of rurality.

## 6. Conclusions

Structural inequities contribute to urban–rural cancer disparities. For rural residents, Structural Urbanism is an example of structural inequities caused by the practice of managing and developing public health services in areas with the highest resource and human capital, consequently funneling resources away from rural areas. Studies report reduced access to and utilization of healthcare services amongst rural cancer patients, but data quantifying differences in cancer support service use remain limited. This study offers important insight regarding urban–rural differences in supportive care use and associated barriers. Specifically, fewer rural patients appear to use secondary support services, and this may be partially due to rural patients’ perceived importance of services. Future research is warranted to confirm these findings with larger, more representative samples and more precise, comprehensive assessments (e.g., medical record documented use of services; use of community and clinic supportive services; validated instruments of perceived importance and other psychosocial factors), and to explore potential avenues for clinical intervention (e.g., culturally appropriate outreach regarding remote support services). In particular, future research should examine the independent and interdependent roles of different determinants (e.g., provider perceptions, patient–provider communication quality, patients’ geographic and economic healthcare access, cultural factors) on urban–rural disparities of patient perspectives and use of secondary support services.

## Figures and Tables

**Table 1 ijerph-19-03405-t001:** Study sample characteristics and crude analyses to assess urban–rural differences in demographic, cancer-related, and severe support needs factors.

	Urban (*n* = 238)		Rural (*n* = 88)		Overall (*n* = 326)		
	*n*	%	*n*	%	*n*	%	*p*-Value
**Rurality**							--
Metro counties with ≥1 million (1)	28	12%			28	9%	
Metro counties with 250,000–1 million (2)	13	5%			13	4%	
Metro counties with <250,000 (3)	197	83%			197	60%	
Counties with ≥20,000, adjacent to a metro area (4)			13	15%	13	4%	
Counties with ≥20,000, not adjacent to a metro area (5)			8	9%	8	3%	
Counties with 2500–19,999, adjacent to a metro area (6)			53	60%	53	16%	
Counties with 2500–19,999, not adjacent to a metro area (7)			13	15%	13	4%	
Counties with <25,000, adjacent to a metro area (8)			0	0%	0	0%	
Counties with <25,000, not adjacent to a metro area (9)			1	1%	1	0.3%	
**Age**							0.32
18–56 years old	66	28%	17	19%	83	26%	
57–65 years old	66	28%	27	31%	93	29%	
66–72 years old	52	22%	25	28%	77	24%	
73+ years old	54	23%	19	22%	73	22%	
**Sex**							0.64
Female	134	56%	47	53%	181	56%	
Male	104	44%	41	47%	145	44%	
**Race**							0.41
Ethnic minority	45	19%	13	15%	58	18%	
Non-Hispanic White	193	81%	74	85%	267	82%	
**Marital status**							0.54
Not married	101	42%	34	39%	135	41%	
Married	137	58%	54	61%	191	59%	
**Insurance status**							0.16
Medicaid	32	13%	10	11%	42	13%	
Medicare	115	48%	53	60%	168	52%	
Private	91	38%	25	28%	116	36%	
**Cancer Dx**							0.55
Non-solid tumors	19	8%	7	8%	26	8%	
Hematopoietic/Lymphoid	37	16%	7	8%	44	13%	
Genital	47	20%	23	26%	70	21%	
Lip/Oral/Pharynx	22	9%	12	14%	34	10%	
Digestive	25	11%	10	11%	35	11%	
Respiratory	35	15%	10	11%	45	14%	
Skin	14	6%	6	7%	20	6%	
Breast	39	16%	13	15%	52	16%	
**Treatment Status**							0.56
Active	103	43%	33	38%	136	42%	
Not active	125	53%	55	63%	190	58%	
**Treatment Type**							
Surgery	157	66%	65	74%	222	68%	0.17
Radiation	87	37%	38	43%	125	38%	0.28
Chemotherapy	129	54%	46	52%	175	54%	0.76
**Multiple Treatments**	145	61%	61	69%	206	63%	0.16
**# of Severe Support Needs** ^†^							0.28
0–1 severe support needs	106	45%	36	41%	142	44%	
2–4 severe support needs	68	29%	23	26%	91	28%	
5+ severe support needs	64	27%	29	33%	93	29%	

^†^ Variable was analyzed continuously but is presented categorically to facilitate interpretability.

**Table 2 ijerph-19-03405-t002:** Crude analyses to assess relationships between demographic, cancer-related, and severe support needs factors with support services use and perceived importance of support services.

	Support Services Use (Yes) *n* = 78			Perceived Importance of Services (Yes) *n* = 198		
	*n*	%	*p*-Value	*n*	%	*p*-Value
**Geographic location**			**0.02**			**0.03**
Urban	65	27%		153	64%	
Rural	13	15%		45	51%	
**Age**			**0.001**			**0.001**
18–56 years old	20	24%		60	72%	
57–65 years old	35	38%		65	70%	
66–72 years old	12	16%		37	48%	
73+ years old	11	15%		37	49%	
**Sex**			0.14			0.37
Female	49	27%		106	59%	
Male	29	20%		92	63%	
**Race**			0.71			0.69
Other	15	26%		34	59%	
Non-Hispanic White	63	24%		164	61%	
**Marital status**			0.85			0.25
Not married	33	24%		87	64%	
Married	45	24%		111	58%	
**Private insurance status**			0.54			**0.006**
Other	48	23%		116	55%	
Private	30	26%		82	71%	
**Reproductive cancer diagnosis**			0.20			0.47
Not reproductive	44	22%		127	62%	
Reproductive	34	28%		71	58%	
**Active treatment status**			**0.001**			**0.009**
Not active	33	17%		104	55%	
Active	45	33%		94	69%	
**Treatment Type**						
Surgery			0.97			0.49
No	25	24%		66	64%	
Yes	53	24%		132	60%	
Radiation			0.28			0.34
No	44	22%		118	59%	
Yes	34	27%		80	64%	
Chemotherapy			**0.03**			**0.05**
No	28	19%		83	55%	
Yes	50	29%		115	66%	
**Multiple Treatments**			**0.001**			**0.04**
No	16	13%		56	47%	
Yes	62	30%		134	65%	
**# of Severe Support Needs** ^†^			0.58			**<0.001**
0–1 severe support needs	34	24%		70	49%	
2–4 severe support needs	19	21%		58	65%	
5+ severe support needs	25	27%		70	75%	

^†^ Variable was analyzed continuously but is presented categorically to facilitate interpretability. **Bold** values have *p* ≤ 0.05.

**Table 3 ijerph-19-03405-t003:** Adjusted logistic regression models examining rurality and supportive service use and perceptions.

Support services Use (None, Any)												
	Model with Age ^1^			Model with			Model with			Final Model ^4^		
				Demographic Factors ^2^			Cancer-related Factors ^3^					
Model Fit	LR	df	*p*-value	LR	df	*p*-value	LR	Df	*p*-value	LR	df	*p*-value
	--	--	--	2.44	4	0.66	16.44	6	0.01	24.93	5	<0.001
	OR	95%CI	*p*-value	OR	95%CI	*p*-value	OR	95%CI	*p*-value	OR	95%CI	*p*-value
Rurality (REF: Urban)	0.48	0.25, 0.92	0.03	0.48	0.25, 0.94	0.03	0.42	0.21, 0.82	0.01	0.43	0.21, 0.85	0.02
Perceived Importance of Support services (None, Any)												
Model Fit	LR	df	*p*-value	LR	df	*p*-value	LR	Df	*p*-value	LR	df	*p*-value
	--	--	--	8.77	4	0.07	20.18	6	0.003	34.88	5	<0.001
	OR	95%CI	*p*-value	OR	95%CI	*p*-value	OR	95%CI	*p*-value	OR	95%CI	*p*-value
Rurality (REF: Urban)	0.60	0.36, 0.99	0.04	0.63	0.38, 1.05	0.08	0.52	0.31, 0.89	0.02	0.55	0.32, 0.94	0.03

Significant results (*p* ≤ 0.05) are marked in bold; LR, Logistic Regression; DF, Degrees of Freedom; OR, Odds Ratio; CI, Confidence interval. ^1^ Age (continuous) included as a covariate. ^2^ Age (continuous), sex (male, female), race (non-Hispanic White, Other), marital status (married, not married), and insurance (private insurance, other) included as covariates. ^3^ Type of cancer diagnosis (reproductive, not reproductive); severe support needs (continuous), surgery (no, yes), radiation (no, yes), chemotherapy (no, yes), multiple treatments (no, yes), and active treatment status (no, yes) included as covariates. ^4^ Age (continuous), insurance (private insurance, other), multiple treatments (no, yes), chemotherapy (no, yes), active treatments (no, yes), and severe support needs (continuous) included as covariates, given these variables were associated with outcomes (see Table 2).

## Data Availability

Restrictions apply to the availability of these data. Data was obtained from a larger cross-sectional needs assessment stored in the Simmons Cancer Institute patient medical records and are available from the authors upon request and with the permission of the Simmons Cancer Institute.

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
