# Peer review of "Consequences of Structural Urbanism: Urban–Rural Differences in Cancer Patients’ Use and Perceived Importance of Supportive Care Services from a 2017–2018 Midwestern Survey"

_ijerph, 2022, doi:10.3390/ijerph19063405_

Round 1

Reviewer 1 Report

There is a burden of unmet cancer care needs among rural cancer patients and their caregivers due to reduced access to cancer care resources.

Structured inequities create a bias in funding for heavily populated health centers and also limit a patient's ability to gather important health information, thereby limiting the perception and understanding of their cancer care needs.

The study analyzed data from a cross-sectional needs assessment about patients' support needs, use of services, and perceptions at a cancer center in the Midwest.

A study packet was provided to cancer patients during their clinic visit, and interested patients completed questionnaires and consented to participate. All data were collected between November 2017 and May 2018. Out of 326 patients, 27% were from rural areas. Among our urban sample, 83% lived in metro counties with <250,000 residents. Among our rural sample, 60% lived in counties with 2,500-19,999 residents that were adjacent to a metropolitan area. Approximately 27% of the sample was rural, 22% was 73+ years old, 56% of the sample were women, 82% were non-Hispanic White, 59% were married, and 36% had private insurance. Approximately 29% of the sample reported 5+ severe support needs. The most common unmet support needs concerned work, weakness, and intimacy.

The analytic Plan is reasonable.

A logistic regression model analyzed adjusted data to determine whether rural versus urban patients reported using secondary support services and perceived secondary support services as very important. Rural patients reported using secondary support services less often (14 vs. 27%; OR = 0.43, 95%CI [0.22, 0.85], p =0.002); and that their perception of secondary support services was lower than that of urban patients (51% vs. 64%; OR = 0.57, 95%CI [0.33, 0.94], p=0.03).

Comments that need to be addressed:

  • Selection bias in choosing patients who speak English and have a residential address- many do not have a mailing address and those who don’t speak English.
  • Another selection bias is gender- 56% women; Patients who used services were more likely to be younger, have undergone chemotherapy, and have multiple treatments.
  • Also, 53-63% had no active status with respect to treatment. There is no mention of the number of patients who did not want to do the survey or did not complete the survey
  • Authors can develop on the special needs of the age group of the patients in the survey- to be mentioned 81% from rural areas that responded were between ages of 57-73+. Additionally, 54-70% were a combination of Medicaid/medicare in urban/rural areas.
  • Since this is a tertiary cancer center, questions around supportive care need education and programming are not provided
  • Also, questions around why the patients decided to come to the tertiary care center are not addressed.

Rural-urban differences in the use of these services should be investigated, given their role in widening disparities in cancer outcomes and their connection with persistent structural inequities.

Reviewer 2 Report

Dear authors, 

thank you let me read your paper. Follows some comments and questions over the paper. I hope I helped you.

1- ABOUT ABSTRACT

The title is too big, it needs to be shortened. In addition, the location and time series in which the study was carried out are missing. Therefore, it is important to insert the time series in the title and in the methods as well.

For example, a short title with full information would be: "Urban-Rural Structural Inequities in Support Services for Cancer Patients and Their Perception: Midwest US - 2020".

“Methods: We used data from a larger cross-sectional needs assessment about patients’ support needs, use of services, and perceptions at a Midwestern United States cancer center. Oncology patients received a study packet during their clinic visit, and interested patients consented and completed the questionnaires”.

Also missing data source, where did you get data? What is the data year? This needs to be included in the methods.

“Conclusion: Our findings suggest links between structural inequities and the access to and importance of supportive care services. To eliminate persistent urban-rural disparities in cancer care, rural residents must have long-term programs and policies that address cancer care and structural inequities

What are the main links between inequalities and the access and importance of services?

Suggest or indicate what type of program or long-term policy? By the way, what is the average duration of a long-term policy? This point should be clear in the conclusion.

2- ABOUT THE TEXT AND CONTENT

“To address this need, a Midwestern sample of urban and rural cancer patients seeking care at the same cancer center was studied. The hypotheses are provided below”.

What is the cancer center? Where is it? Why is this cancer center important? Why did you choose this cancer center?

I saw that you got data from “cancer center affiliated with a university in the Midwestern United States”, but you need to respond the questions above.

However, the article is very good.

Wish best wishes
